# The Importance of MDP Priming, Silica Blasting or Glazing on the Retention Force of Y-TZP Copings to Varying Geometry Tooth Abutments

**Nathália C. Ramos** [1,*], **Larissa M.M. Alves** [1], **Gabriela F. Ramos** [1], **Marco Antonio Bottino** [1], **Renata M. Melo** [1] and **Rodrigo Othávio A. Souza** [2]

[1] Department of Dental Materials and Prosthodontics, Institute of Science and Technology, São Paulo State University (UNESP), 777 Eng. Francisco José Longo Avenue,
São José dos Campos 12245-000, São Paulo, Brazil; larissammalves@gmail.com (L.M.M.A.);
gabrieladsfreitas@gmail.com (G.F.R.); mmbottino@uol.com.br (M.A.B.);
renata.marinho@ict.unesp.br (R.M.M.)

[2] Dentistry Department, Federal University of Rio Grande do Norte (UFRN), 1787 Salgado Filho Avenue,
Natal 59056-000, Rio Grande do Norte, Brazil; rodrigoothavio@gmail.com

\* Correspondence: nathalia.carvalhoramos@gmail.com; Tel.: +55-12-39479032

**Abstract:** To evaluate the influence of the convergence angle of tooth preparations and abutments height and several surface treatments for zirconia copings through the tensile retention test. 120 crown preparations were made in Nema G10 with the maxillary first molar anatomy. In total, 60 abutments of 5 mm height were divided into two groups of 6° and 20° convergence angles of tooth preparations, and 60 abutments with a convergence angle of tooth preparations of 12° were divided into groups of 4 and 6 mm heights. Three surface treatments used were MDP-primer (10-Methacryloyloxydecyl dihydrogen phosphate), glazing or silica blasting. The abutments were scanned to make zirconia copings (3Y-TZP–Yttria-Stabilized Tetragonal Zirconia Polycrystals, Vita In-Ceram YZ). After cementation, the mechanical cycling ($2 \times 10^6$ cycles, 3 Hz, 100 N) was performed to aging. After cycling, the copings were tested in tensile (1 kN load cell; 0.5 mm/s speed). Both abutments support base and copings were embedded in acrylic resin with the aid of a device that maintained the long axis perpendicular to the horizontal plane. Data were analyzed with the two-way ANOVA and Tukey test (95%). ANOVA revealed that the convergence angle influenced the tensile retention ($p = 0.0232$), but the abutments height showed no statistically significant difference ($p = 0.086$). The MDP-primer and silica blasting showed higher retention forces in the specimens with height variations. For bonded zirconia crowns, the retention force provided by high convergence angle preparation is critical and cannot be improved by surface treatments. For short and long crown preparations, MDP-based Primers or Silica blasting are advisable to aid restoration longevity.

**Keywords:** Y-TZP ceramic; tensile strength; tooth preparation; dental bonding; dental prosthesis failures

## 1. Introduction

Currently, all-ceramic restorations are widely used mainly because they meet the aesthetic and strength needs. As much as monolithic restorations represent a trend, bilayer restorations are still a more favorable aesthetic solution and are associated with greater mechanical strength. The use of porcelain-covered zirconia copings is a combination that can produce all-ceramic restorations with higher repairable failure rates compared to catastrophic failures with higher cumulative survival rates [1,2].

The clinical success of indirect restorations is linked to the ability of the dental professional and the dental technician, besides the materials behavior and others factors. In spite of that, further information on how much the convergence angle or height of the tooth preparations will affect the performance of adhesively bonded zirconia are necessary [3,4].

Some studies have shown that the convergence angle and height of the tooth preparation can influence the resistance to debonding. The convergence angle is a gradual decrease in the width of the preparation, whereas a smaller convergence angle is directly related to adequate strength and retention [5–8].

Preparations with zero convergence degree inclination of their walls (fully parallel walls) have better retention, but render crown removal difficult due to frictional forces. According to some authors, the ideal taper of a preparation is around 6° [4,7]. However, to obtain restoration stability, other studies showed that the convergence angle, varying from 12° to 25°, is sufficient [6,7,9].

Another important factor is whether the height of preparation influences the retention and rotational stability of the future prosthesis [10]. In addition to the geometry of tooth preparation, bonding agents for cementation are necessary for longevity [4]. In cases of atypical preparation or low-retention preparations (worst case scenario), the dentist needs to use surface treatments in the internal parts of the copings to achieve greater adhesion of the restoration to the dental substrate.

Various surface treatments are performed to improve the performance and longevity of such all-ceramic restorations [11]. The silica blasting technique is used to change the microretentive surface, increasing amounts of zirconia bonding to the resin, but some studies have shown that this technique can reduce the mechanical properties of ceramics due to the high impact of the particles of silica on zirconia, thus inducing crack propagation [11,12]. Another method is silanization, which provides a chemical bond between the ceramic inorganic phase and the resin-based organic phase applied to the etched ceramic surface, but as Y-TZP (Yttria-Stabilized Tetragonal Zirconia Polycrystals) cannot be etched, primers containing phosphate-based monomers to promote greater adhesion can also be used [13]. Multi-mode or universal adhesives have appeared, which also have MDP (10-Methacryloyloxydecyl dihydrogen phosphate) and silane in its composition, but Araujo et al. (2018) showed that this type of universal adhesives did not improve the zirconia bonding [14]. A surface treatment technique more recently used is the formation of a vitreous layer on the surface, allowing the zirconia to be treated by acid and silane [15]. This is achieved by glazing the zirconia surface, which was shown to increase the ability of bonding with resin cements [16,17].

Surface treatments are extremely necessary not only to ensure the adhesion between zirconia and resin cement and dental substrate- but also to perform an important role in the material behavior. Surface coatings can act on the surface roughness and bacterial colonization, affect chemical solubility and/or the pH of the oral environment, promote or fill defects in the material that would be crack initiators, and also alter the wear coefficient, among other factors [18].

Zirconia crowns have a stress peak and average stress higher than any other ceramic, thus, the study by Dal Piva et al. (2018) showed that the failure risk is 0.20 and 0.09 for cohesive and adhesive failures, respectively. These data show the need to further study the zirconia behavior under tensile stresses, such as the tensile retention test used in this work [19].

Thus, the aim of the present study was to evaluate the influence of the convergence angle and height of preparations with several surface treatments on the tensile retention force of Y-TZP ceramic copings, there by testing the hypothesis that differences would arise due to the convergence angle, height, and surface treatment variations. The null hypothesis was that there were no differences among the convergence angle, height, or surface treatment variations.

## 2. Materials and Methods

### 2.1. Preparations of Specimens

In total, 120 anatomical replicas based on abutments of first human molar anatomy were fabricated using an epoxy resin reinforced by glass (NEMA G-10, International Paper, Hampton, TN, USA), with elastic properties similar to dentin [20]. The study design is

presented through the schematic figure (Figure 1); the sample size was based on the study by Amaral et al. (2014) [21]. The preparations were divided according to convergence angle and height of the preparations, as seen in Figure 2, which were then further subdivided into surface treatments: MDP-primer, silica blasting and glazing that are described above.

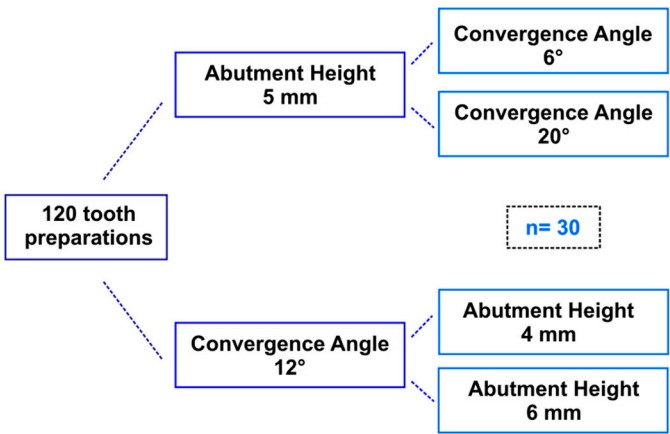

**Figure 1.** Schematic representation of the study design.

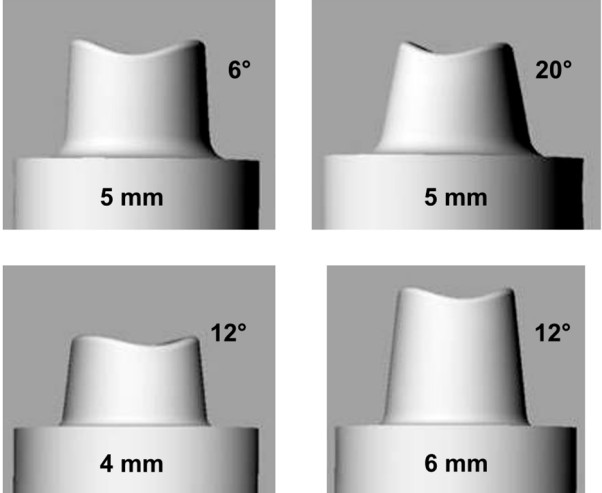

**Figure 2.** Representation of the preparations design. The groups with 6° and 20° of convergence angle tooth preparation have 5 mm height, while the groups with 4 and 6 mm of height have 12° convergence angle.

The preparations were embedded in a polyvinyl chloride-PVC tube using acrylic resin; the occlusal surface was positioned parallel to the horizontal plane with the help of an adapted device. The preparations were scanned, and a three-dimensional image was obtained in InLab 4.0 software (Sirona Dental Systems GmbH, Bensheim, Germany). The copings were designed with retentions in the external surfaces so that retention to acrylic resin was not lost, as seen in Figure 3. The copings were milled in pre-sintered blocks of 3Y-TZP (Vita In-Ceram YZ, Vita Zahnfabrik, Bad Säckingen, Germany) in the in Lab MC XL CEREC® milling machine (Sirona Dental Systems GmbH) with internal relief of 70 μm. The copings were then sintered in a Vita Zyrcomat furnace (Vita Zahnfabrik, Bad Säckingen, Germany).

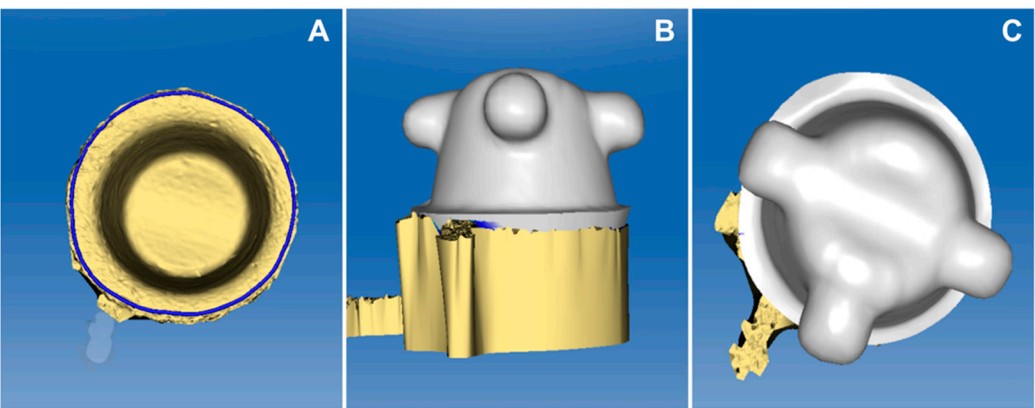

**Figure 3.** The images represent a CAD (computer-aided design) process: (**A**) delimitation of the cervical edge of the preparation; (**B**) preparation coping design (front view); and (**C**) preparation coping design (upper view).

### 2.2. Surface Treatments

After sintering, the copings were cleaned in two ultrasonic bath with isopropyl alcohol and distilled water, and then received one of the following surface treatments:

(1) MDP-primer: A layer of Monobond N (Ivoclar Vivadent, Schaan, Liechtenstein) was applied on the internal surface of the coping with a microbrush, leaving it undisturbed for five min before cementation.

(2) Silica blasting: Sandblasting of zirconia with aluminum oxide particles coated with silica (30 μm) (Cojet Sand - 3M ESPE, St. Paul, MN, USA), using a sandblasting device (Dento-Prep-Rønvig Dental Mfg. A/S, Daugaard, Denmark). The distance between the ceramic surface and the tip of the device was 10 mm with a 45° inclination angle on a metallic support. The pressure was 2.5 bars and blasting time was 10 s. The silane MDP-free agent was Monobond S (Ivoclar Vivadent, Schaan, Liechtenstein), and was applied for 5 min before cementation.

(3) Glazing: The glaze layer (Vita Akzent glaze, Vita Zahnfabrik, Bad Säckingen, Germany) was manually applied with the assistance of a brush and sintered according to the firing protocol: Washburn sintering, initial temperature, 500 °C; heating time, 5 min; temperature elevation rate, 80 °C/min; final temperature, 900 °C- and setting time at the final temperature, 5 min. Prior to cementation, the glazed surface was etched with 10% hydrofluoric acid (HF) for 1 min [17], washed with water spray for 2 min, and then the coping was subjected to an ultrasonic bath for 5 min to remove the precipitates. Monobond S silane (Ivoclar Vivadent, Schaan, Liechtenstein) was then used, acting for 5 min before cementation.

### 2.3. Luting Procedures

After complete evaporation of the silane, the copings were randomized inside the respective group, then cemented with dual-curing resin cement (Variolink II, IvoclarVivadent, Schaan, Liechtenstein), in which a base and catalyst were applied in equal proportions. The cement was applied on the inner side walls of the coping, which was brought into position and kept under a constant load of 750 g. The assembly was cured for 40 s on the buccal, lingual and occlusal sides (total time 120 s, light intensity 1200 mW/cm$^2$) (Radii-cal, SDI, Bayswater, Victoria, Australia). The same operator performed all surface treatments, and another operator (blind regarding the surface treatments) performed the luting procedures.

### 2.4. Fatigue and Tensile Test

After cementation, the samples were stored in distilled water at 37 °C for 24 h to ensure the complete polymerization of the resin cement. They were then placed in a mechanical cycling machine (Biocycle V1, BioPdi, São Carlos, Brazil) with an axial load of 100 N on the occlusal surface center and a frequency of 3 Hz for $2 \times 10^6$ cycles to mechanical aging, also

in distilled water. To diminish the impact between the indenter and the occlusal surface, a silicon film (1 mm thick) was positioned on the top surface of the coping. No failures were found during or at the end of the fatigue aging test.

After aging, the copings were also embedded in acrylic resin with the aid of a device that maintained the long axis perpendicular to the horizontal plane. The specimens inclusion and the tensile test followed the study of Amaral et al. (2014) [21]. The specimens were subjected to a tensile test in a universal testing machine (EMIC DL 1000, São José dos Pinhais, Brazil) at a crosshead speed of 0.5 mm/min using a load cell of 1 kN. A device containing two universal joints was used to hold the copings and ensure the occurrence of uniaxial tensile forces and to avoid lateral forces. The maximum load to debond the coping from substrate was recorded (in N), which denotes the tensile force of the specimen.

*2.5. Failure Analysis*

Specimens were evaluated with a stereomicroscope (Discovery V20, Carl-Zeiss, Gotingen, Germany), with the purpose of evaluating the type of fracture. The failures were divided into three forms: G10 adhesive (GA) (50% or more cement was in the coping), zirconia adhesive (50% or more cement was in the G10 surface), and mixed adhesive (when there was resin cement in the coping and G10 surface).

*2.6. Statistical Analysis*

Two-way ANOVA and the Tukey's test were applied to analyze the results of tensile retention force of Y-TZP ceramic copings (Minitab 17, Minitab Inc., State College, PA, USA).

**3. Results**

The samples with a convergence angle of 6° showed higher tensile retention force than the samples with 20°. ANOVA revealed that the "convergence angle" factor (6° and 20°) was significantly different ($p = 0.0232$), but the "surface treatment" and "interaction of convergence angle and surface treatment" factor were not ($p = 0.455$, $p = 0.6909$), respectively. The means, standard deviations, and the results of the Tukey test, comparing the surface treatments and convergence angles, are shown in Table 1.

**Table 1.** Mean (N), standard deviation and homogeneous groups (Tukey test, $p < 0.05$) comparing the surface treatments and convergence angles and comparing the surface treatments and preparation heights.

| | Tensile Retention Force Mean (SD) | | | |
|---|---|---|---|---|
| _ | Convergence Angle (5 mm of Height) | | Height (12° of Convergence Angle) | |
| _ | 6° | 20° | 4 mm | 6 mm |
| Silica Blasting | 667.7 ± 190.4 [a] | 496.6 ± 150.2 [a] | 593.1 ± 198.5 [AB] | 599.6 ± 150.5 [AB] |
| MDP-Primer | 658.9 ± 77.50 [a] | 491.6 ± 164.2 [a] | 467.4 ± 168.8 [AB] | 705.0 ± 162.9 [A] |
| Glaze | 557.9 ± 256.3 [a] | 446.2 ± 151.7 [a] | 443.2 ± 205.7 [B] | 464.3 ± 243.2 [AB] |

* Equal letters mean no statistically significant differences and different superscript letters mean statistically significant differences among the groups.

Considering the factor "height", there was no statistically significant difference between them ($p = 0.086$), but the surface treatments were significantly different ($p = 0.045$), with best performance of the MDP-primer. The means, standard deviations and the results of the Tukey test, comparing the surface treatments and the preparation heights are shown in Table 1.

The specimens were analyzed in a stereomicroscope and showed the three previously mentioned failure types. The percent of each failure and the examples of the failure types are shown in the Table 2 and Figure 4, respectively.

**Table 2.** Percentage of failures according to the following classification: adhesive substrate (when 50% or more cement was in the coping), adhesive zirconia (when 50% or more cement was in the G10 surface) and adhesive mixed (when there was resin cement in the coping and G10 surface).

|  |  | Adhesive Substrate | Adhesive Zirconia | Adhesive Mixed |
|---|---|---|---|---|
| **6°** | Silica blasting | 90% | – | 10% |
|  | MDP-primer | 90% | – | 10% |
|  | Glazing | 90% | – | 10% |
| **20°** | Silica blasting | 100% | – | – |
|  | MDP-primer | 100% | – | – |
|  | Glazing | 50% | 50% | – |
| **4 mm** | Silica blasting | 100% | – | – |
|  | MDP-primer | 90% | – | 10% |
|  | Glazing | 40% | 50% | 10% |
| **6 mm** | Silica blasting | 100% | – | – |
|  | MDP-primer | 80% | 10% | 10% |
|  | Glazing | 20% | 80% | – |

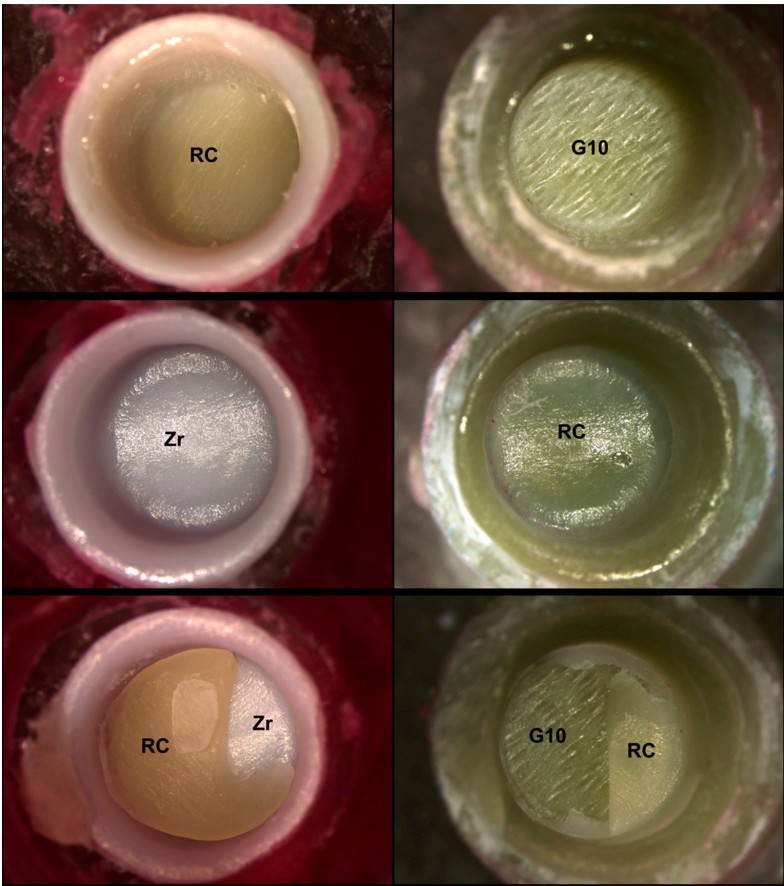

**Figure 4.** Stereomicroscope images–10 × objective. On the right side are the G10 preparations, and the copings made from zirconia are on the left side. The first row shows a "G10 adhesive failure", where the resin cement (RC) is on zirconia coping. In contrast, the second row shows a "zirconia adhesive failure" pattern, where the resin cement is on the G10 surface. The third row presents a "mixed (cohesive and adhesive) failure", where resin cement can be observed on both surfaces (zirconia coping and G10 substrate).

## 4. Discussion

The abutments design play a significant role in crown retention. In this study, we aimed to evaluate the heights and convergence angles of preparations for adhesively-bonded zirconia crowns. The null hypothesis could not be confirmed, because the groups with convergence angle variation presented no statistically significant difference, regardless of the surface treatment. That is, all the surface treatments were similar in both convergence angle conditions (12° and 20°). The groups with abutment height variation presented statistical difference depending on the surface treatment. However, all these groups were similar, except the group that was glaze-treated with a 4 mm abutment height and MDP-primer with a height of 6 mm, which are different from each other.

The physical aspect was predominant on tensile retention of Y-TZP-copings- when there was a variation of the convergence angle, showing that the lowest preparation angle presented the best tensile retention in accordance with the authors Shillingburg [8] and Goodacre et al. [22].

In the present study, the height of preparation did not influence the tensile retention of the coping; this fact may occur because the heights of the preparations were greater than 3 mm (4 and 6 mm). Leong et al. showed that only a height less than 3 mm influences the resistance and retention of the preparation [10]. However, the surface treatment presented statistical differences depending on the height, and MDP-primer was shown to be the best option for increasing the tensile retention when compared to glazing, since it provides direct interaction with the zirconia surface, promoting the adhesion between the materials [11,23]. The MDP acts on the hydroxyl groups of the zirconia oxide surface layer, interacting at an interfacial level. Thus, each phosphate group joins on to one or three zirconium atoms, creating phosphate zirconium, which is a more thermally and hydrolytically stable zirconium salt, which may justify the better adhesive behavior of this surface treatment [24].

Despite the fact that the "glaze on technique" is considered an innovative technique, there are still few studies that show its efficiency [16,17]. A low tensile retention of glazed samples can be explained by the application technique used in the present study, resulting in a heterogeneous layer, which may have contributed to bonding instability and low retention force [16,25].

A retrospective study of implant-supported zirconia bridges (porcelain-covered) showed a cumulative survival rate of 100% after 5 years, and the failures found were only about the porcelain chipping [2]. In this study, most failures were at the adhesive interface between cement/G10, showing that the surface treatments were effective on the zirconia copings, which is in accordance with other studies [4,23,26]. Silica blasting and the MDP-primer are surface treatment that were well stabilized in the literature, showing higher bond strength values [26,27]. One exception was the glazing groups that also presented failures at the zirconia/cement interface. This shows that this technique is not as predictable as priming or silicatization for bonding to zirconia. Another study has shown the same trend [17].

In fact, none of the surface treatments used herein were able to compensate for the loss of parallelism of the preparations. Literature about bonding to zirconia showed that the association of silicatization and MDP-based primers resulted in higher bond strengths than both treatments made separately, warranting more studies on the effect of surface treatments and different convergence angles [11,28–30].

Higher retention force can also be related to the internal adaptation of the copings. In one study [31], preparations with 20° of taper showed smaller internal spaces, which was different for the group with a convergence angle of 6°. These findings would indicate that the internal space is reduced as the convergence angles of the abutments increase. This is probably a result of reduce layer cement thickness due to its improved flow rate at higher convergence angles. This improvement in cement flow and crown seating did not have an effect in the present study. On the other hand, glazing on the internal surface may have affected the internal adaptation, decreasing the retention force of the short and long preparations.

The use of NEMA G10 substrate is well established in the literature, according to its elastic modulus compatibility and the fact that when acid etched, the material's fibers are exposed, creating micro retentions where the resin cement fills. Thus, bond strength and mechanical simulations using this epoxy resin is similar to hydrated dentin [20]. Besides that, periodontal ligament simulation and the presence of dental roots are not necessary to guarantee the same stress distribution that occurs in the tooth [32]. In addition, aging conditions simulate the mechanical fatigue that occurs in an oral environment, but an in vitro study, it still has limitations. Among the limitations, the most important concerns the tensile force test, which does not represent the stress pattern to which the tooth is submitted under masticatory loads. Besides that, limitations of this study are the disuse of the combinations of air-abrasion and the MDP-primer, and the other glaze applications techniques as well. Future studies should be based on the association of surface treatment techniques with worst case scenarios such as a high convergence angle with short preparations.

## 5. Conclusions

For bonded zirconia crowns, the retention force provided by high convergence angle preparation is critical and cannot be improved by surface treatments. For short and long preparations, MDP-based primers or silica abrasion are advisable as surface treatment to aid the longevity of the restoration adhesion.

**Author Contributions:** Conceptualization, R.O.A.S. and R.M.M.; methodology, N.C.R., G.F.R., and L.M.M.A.; software, G.F.R.; validation, N.C.R., R.M.M., and M.A.B.; formal analysis, N.C.R. and G.F.R.; investigation, L.M.M.A.; resources, M.A.B.; data curation, N.C.R.; writing—original draft preparation, N.C.R. and G.F.R.; writing—review and editing, N.C.R., L.M.M.A., G.F.R., R.O.A.S., and R.M.M.; visualization, L.M.M.A.; supervision, R.O.A.S. and M.A.B.; project administration, R.M.M.; funding acquisition, M.A.B. All authors have read and agreed to the published version of the manuscript.

**Funding:** This research received no external funding.

**Data Availability Statement:** All data included in this study are available upon request by contact with the corresponding author.

**Acknowledgments:** We acknowledge that Evelyn B.C. Monteiro Dall'acqua and Carolina S. Almeida gave technical contributions to the development of the present study. This research did not receive any specific grant from funding agencies in the public, commercial, or not-for-profit sectors.

**Conflicts of Interest:** The funders had no role in the design of the study; in the collection, analyses, or interpretation of data; in the writing of the manuscript, or in the decision to publish the results.

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
