# Peer review of "The Importance of MDP Priming, Silica Blasting or Glazing on the Retention Force of Y-TZP Copings to Varying Geometry Tooth Abutments"

_coatings, doi:10.3390/coatings11030315_

Round 1

Reviewer 1 Report

The present study aimed to evaluate the influence of the convergence angle, abutments height and several surface treatments for zirconia copings through the tensile retention test. The subject is interesting, fits the special issue’s scope and can contribute to the scientific literature. However, minor corrections are necessary in order to improve the text quality.

Title:

Rewrite your title giving more focus to the surface treatments.

Abstract:

Please use “convergence angle of a tooth preparation” instead just “convergence angle”;

In all text, separate the unity from the numeral e.g. “5mm”’ to “5 mm”;

Correct the number of cycles in the aging parameter;

Describe how the samples were fixed;

Insert the universal testing machine reference with load cell and speed;

In your last sentence, use “crown preparation” instead just “preparation”.

Introduction:

In your first paragraph, describe the use of zirconia copings in dental bilayer restorations, explaining the importance of this material as a framework.

In your introduction section, with a proper reference, explain if the application of multi-mode adhesive could be a substitute to silanized Y-TZP ceramics.

Insert the importance to study tensile resistance in adhesively bonded zirconia restorations. What is the correlation between tensile stress and zirconia crowns?

Describe the importance of dental materials coatings and surface treatments in order to improve its adhesion.

Methods:

Please, insert the reference that have validated the use of NEMA G10 as dentin substrate substitute for in vitro mechanical tests;

The authors believe that different preparation designs will modify the results? What is the influence of substrate design for in vitro mechanical testing?

Describe the samples embedding procedure;

For the Vita Zyrcomat furnace, insert the manufacturer’s complete information;

Describe how the samples were cleaned before the surface treatments;

For the Monobond S, insert the manufacturer’s complete information;

The glaze layer internal application is very interesting. Is there a reference for the applied surface treatment with HF on it?

For the resin cement, insert the manufacturer’s complete information;

The light source intensity should be corrected;

Describe why the samples were stored in distilled water at 37ËšC for 24 hours;

Correct the number of cycles in the aging process;

Results:

Replace “better tensile retention “for “higher tensile retention”;

In table 1, remove the parentheses and use ± to separate average and standard deviation;

Correct “between then” to “between them”;

Remove the failures type description from results, it was already explained in the methods section.

In table 2, replace “Adhesive G10” to “Adhesive Substrate”;

Discussion:

Correct “Our results showed and the null hypothesis”;

Replace “no matter adhesive treatment” to “regardless the surface treatment”;

In second paragraph, please use “showing that the lowest preparation angle presented” instead “showing that the lowest presented”;

Correct the reference citation in the text, in the following sentence: “in accordance with the authors Shillingburg and Goodacre et al.”

Discuss how the primer chemically interacts with the zirconia surface;

In the sentence “possibly debonding of the crowns”, I will suggest the use of “low retention force”, since debonding occurred in all samples and was purposed occasioned by the test compliance.

Replace “low cement pellicle thickness” to “reduce cement layer thickness”

Describe your study’s limitations in the last paragraph.

Conclusion:

In your conclusion, use:

“For short and long preparations, MDP-based Primers or silica abrasion are advisable as surface treatments to aid the longevity of the restoration adhesion.”

Reviewer 2 Report

I've reviewed the manuscript titled "Influence of convergence angle, height and surface treatments on the tensile retention of the Y-TZP copings". Please find my comments bellow:

  1. Line 43, "clinical success linked to...." authors should elaborate several other factors in addition to dentist factor on clinical success
  2. Fig 1, it is advised to modified the representation of the preparations design by keeping convergence angle in one row and height in an other for better and easy understanding
  3. Would you please explain why Monobond S was applied in place of Monobond N in Silica Blasting and Glazing groups? How would it be benefit without Monobond S silane in each group?
  4. Line 167, may be a typing mistake "between then"
  5. Line 130,Fatigue and tensile test methodology is not clear, also describe how did you calculated failure stress or tensile stress, especially Table 1, 
  6. Also from Table 2, most failures are Adhesive G10, please discuss the reasons and why do you think there is role or surface treatment here? 
  7. Please provide further discussion about the epoxy resin reinforced by glass (NEMA G-10) used as tooth replica, as resin cement may bond  chemically with the resin unlike dentine .....
  8. Please describe and discuss results further especially line 192, "presented statistically significant differences, no matter adhesive treatment", 
  9. Lastly, please discuss on the association of surface treatment techniques in Zn if there is any in the present study

Reviewer 3 Report

Study design must to be reported in the title.

Lines 42-43 "The clinical success of indirect restorations is directly linked to the ability of the dental professional.". I disagree. The clinical success of indirect restorations depend by several factors, including but not limiting to the ability of the dental team (dentist and dental technician). For the latter...

Methods

Please check the CRIS guidelines.

Study design, location and other general information must to be reported early in the M&M.

Sample size calculation must to be declared.

Please explain in detail how the samples were fabricated and how the samples were allocated to the groups (surface treatments).

Who perform the treatment? Please define.

Who perform luting procedures? And the tests?

Were the samples randomized? Were the outcome assessors blinded? Define.

Results

Please clearly define all the measures, including “interaction” factor" in the material&methods.

Delete "ANOVA showed that" (line 156) and replace "ANOVA revealed" with "there was" (line 166)

Discussion

The preparation designs... please clarify The abutment-design...

Consider the following paper for the discussion:

1.Pozzi, A., Holst, S., Fabbri, G. & Tallarico, M. Clinical Reliability of CAD/CAM Cross-Arch Zirconia Bridges on Immediately Loaded Implants Placed with Computer-Assisted/Template-Guided Surgery: A Retrospective Study with a Follow-Up between 3 and 5 Years. Clinical Implant Dentistry and Related Research 17 Suppl 1, e86–e96 (2015).

Reviewer 4 Report

The paper concerns interesting dental rehabilitation problems. The study appears well conducted and analyzes the influence of convergence angle, height and surface treatments on the tensile retention of the Y-TZP copings.

However, I must highlight that the paper doesn't bring any significant changes to current knowledge on these topics.

The introduction should be expanded and revised.

The discussion should be rewritten and supplemented by citing recent studies that support the validity of the measurements. 

References: the cited textbooks should be replaced with recent articles; in the reference the authors should report some recent studies to strenghten the scientific validity of the paper.

Author Response

Dear Reviewer 4, 
Thank you for the time taken in our manuscript. The authors really appreciate your time spent and your considerations. The manuscript was improved after this review and we hope that you can consider the new arguments. The corrections/inputs were highlighted in red for better tracking. About the references, there are some old references (around 1980) for replace the book text, but we prefer to keep the recent reference as possible. And we didn’t find others papers that support the tooth preparations for all-ceramics. However, others references were added in the manuscript.

Round 2

Reviewer 4 Report

I appreciate the corrections and the inputs of the authors. The paper may be accepted for publication in its present form